# Surface Coordination of Pd/ZnIn_2_S_4_ toward Enhanced Photocatalytic Activity for Pyridine Denitrification

**DOI:** 10.3390/molecules28010282

**Published:** 2022-12-29

**Authors:** Deling Wang, Erda Zhan, Shihui Wang, Xiyao Liu, Guiyang Yan, Lu Chen, Xuxu Wang

**Affiliations:** 1State Key Laboratory of Photocatalysis on Energy and Environment, Fuzhou University, Fuzhou 350002, China; 2Province University Key Laboratory of Green Energy and Environment Catalysis, Ningde Normal University, Ningde 352100, China; 3Fujian Provincial Key Laboratory of Featured Materials in Biochemical Industry, Ningde Normal University, Ningde 352100, China

**Keywords:** ZnIn_2_S_4_, surface coordination, denitrification, pyridine, photocatalysis

## Abstract

New surface coordination photocatalytic systems that are inspired by natural photosynthesis have significant potential to boost fuel denitrification. Despite this, the direct synthesis of efficient surface coordination photocatalysts remains a major challenge. Herein, it is verified that a coordination photocatalyst can be constructed by coupling Pd and CTAB-modified ZnIn_2_S_4_ semiconductors. The optimized Pd/ZnIn_2_S_4_ showed a superior degradation rate of 81% for fuel denitrification within 240 min, which was 2.25 times higher than that of ZnIn_2_S_4_. From the in situ FTIR and XPS spectra of 1% Pd/ZnIn_2_S_4_ before and after pyridine adsorption, we find that pyridine can be selectively adsorbed and form Zn⋅⋅⋅C-N or In⋅⋅⋅C-N on the surface of Pd/ZnIn_2_S_4_. Meanwhile, the superior electrical conductivity of Pd can be combined with ZnIn_2_S_4_ to promote photocatalytic denitrification. This work also explains the surface/interface coordination effect of metal/nanosheets at the molecular level, playing an important role in photocatalytic fuel denitrification.

## 1. Introduction

Nitrogen-containing compounds (NCCs) have been regarded as some of the most significant atmospheric pollutants, whose combustion will cause serious environmental pollution [1,2,3] and the excess release of NOx into the air, causing acid rain [4]. Petroleum contains high amounts of NCCs [5,6,7,8], such as pyridine. Pyridine, as a representative nitrogen heterocycle, is reported to exert toxic, teratogenic, and carcinogenic effects, and it is classified as a priority pollutant [9,10]. Meanwhile, the stable structures of pyridine are difficult to degrade and they persist for long periods in the environment [11,12]. Therefore, the removal of pyridine from crude fuels has become a global research hotspot [13,14,15]. The question of how to deal with these problems has attracted the attention of researchers.

Thus far, the main denitrification method is HDN, which is expensive and inefficient because of its strict temperature and pressure requirements [16,17]. Alternative technologies have been studied, such as absorption [18], ODN [19], and photocatalysis [20]. Among them, photocatalytic technology has been regarded as a prospective technology for converting NCCs into fuels because of its good selectivity, the direct utilization of sunlight at room temperature, and high efficiency [20,21,22], which is consistent with the concepts of environmental protection and sustainability. Moreover, oxidation processes play an important role in photocatalytic pyridine denitrification. It was reported that pyridine was attacked by free radicals or adsorbed photons, leading to ring opening and causing a range of oxygen-containing products in the process of photocatalysis [15,23]. Meanwhile, a study found that the pyridine denitrification efficiency has been restricted by the rapid recombination of photogenerated carriers, insufficient reactive sites, and sluggish kinetics of the electron transfer process [24]. Therefore, it is desirable to design a photocatalyst with superior electron transfer and separation.

Two-dimensional nanosheets not only possess a high specific surface area and abundant catalytically active sites but also shorten the diffusion distance of charge carriers [25]. Notably, ZnIn_2_S_4_, with a 2D structure, serves as visible light response photocatalyst with a distinctive and tunable electronic structure for improved photocatalytic performance [26,27]. Consequently, ZnIn_2_S_4_ photocatalysts have aroused attention in photocatalytic hydrogen production [28], CO_2_ reduction [29], and contaminant degradation [30,31,32,33]. However, for prospective uses in the actual world, ZnIn_2_S_4_’s photocatalytic activity needs to be substantially enhanced [34,35]. It is well known that semiconductor photocatalysts’ photocatalytic activity can be significantly increased by coupling cocatalysts [36,37,38,39,40,41,42]. In current photocatalytic applications, noble metals such as Pt, Au, and Pd are commonly used because of their ability to form Schottky barriers, which can accelerate the separation of photogenerated carriers [43]. Moreover, compared with other noble metals, Pd has a relatively high Fermi energy level, low electronic affinity for effective electron transfer to protons, and a high state density close to the semiconductor Fermi energy level [44]. Meanwhile, numerous studies have shown that the use of surfactants can significantly improve the dispersibility and contact area of semiconductors, further improving the reaction efficiency [45,46,47,48,49].

Herein, Pd/ZnIn_2_S_4_ heterostructures were successfully synthesized for the selective transformation of pyridine under visible light. Systematic studies showed that the 1% Pd/ZnIn_2_S_4_ heterojunction exhibited the optimal degradation rate of 81% in the photocatalytic denitrification of pyridine, being nearly 2.25 times higher than that of pristine ZnIn_2_S_4_. Such remarkably enhanced photocatalytic performance was mainly ascribed to the Zn⋅⋅⋅C-N or In⋅⋅⋅C-N coordination for pyridine molecules’ selective adsorption and activation. Meanwhile, the electron separation and transfer ability of Pd, as well as SPR, affect Pd. Under visible light irradiation, the Pd/ZnIn_2_S_4_ composites outperform pure ZnIn_2_S_4_ in photocatalytic performance and cycle stability, confirming that the material has excellent fuel denitrification performance. Finally, we suggest a potential photocatalytic mechanism for the degradation of pyridine by Pd/ZnIn_2_S_4_ at the molecular level.

## 2. Results

### 2.1. Characterization

The crystal structure of the synthesized photocatalyst was characterized by XRD. As shown in Figure 1, all characteristic peaks of the pure ZnIn_2_S_4_ (ZIS) were consistent with the ZnIn_2_S_4_ hexagonal. The diffraction peaks located at 27.7°, 30.5°, and 47.2° were ascribed to the (102), (104), and (110) crystal planes of hexagonal ZnIn_2_S_4_ (PDF: 65-2023) [29]. However, due to the low quantity and homogeneous dispersion of Pd, the distinctive diffraction peaks of Pd were not observed.

The morphologies and microstructures of ZIS and Pd/ZIS were characterized by SEM and TEM. As displayed in Figure 2a, the synthesized ZIS exhibited a flower-like structure with a diameter of 4–5 μm, which was composed of abundant 2D nanosheets. After Pd was loaded on ZIS, the flower-like structure was well maintained (Figure 2b). As shown in Figure 2c,d, the Pd nanoparticles had a uniform distribution on the ZIS ultrathin nanosheet surface. In addition, the magnified HRTEM images shown in Figure 2d display obvious lattice fringes of ca. to 0.32 and 0.22 nm, which correspond to the (102) crystal plane of hexagonal ZIS and the (111) plane of face-centered cubic Pd [36]. It is further confirmed that the development of a 2D nano-junction between Pd and ZIS indicates intimate contact between them. Meanwhile, the EDX spectrum of the Pd/ZIS heterojunction demonstrates the presence of Zn, In, S, and Pd elements in Pd/ZIS composites (Figure 2e). In addition, the actual proportion of Pd in the 1% Pd/ZIS photocatalyst is shown in Appendix A. According to the above results, it is indicated that the Pd nanoparticles were successfully uniformly dispersed on the ZIS surface.

The surface chemical states of the as-prepared samples were investigated by XPS analysis. The binding energies in the XPS spectra were calibrated by the adventitious carbon C 1s peak at 284.8 eV. As shown in Figure 3, four elements, Zn, In, S, and Pd, can be observed, which are consistent with the EDS results. The binding energy of Zn 2p can be resolved into two peaks located at 1044.48 eV and 1021.45 eV, which correspond to Zn 2p_1/2_ and Zn 2p_3/2_, respectively [32]. As shown in Figure 3b, the two peaks exhibited at 451.99 and 444.45 eV could be assigned to In 3d_3/2_ and In 3d_5/2_ [39]. Meanwhile, the binding energies of S 2p can be divided into two peaks located at 162.54 and 161.25 eV, which were attributed to the S 2p_1/2_ and S 2p_3/2_ [43]. Compared with the ZIS, the Zn 2p, In 3d, and S 2p of 1% Pd/ZIS exhibited a 0.4 eV blue shift, which could be ascribed to the electron transfer between the two components [40]. In addition, the peaks at 341.10 and 335.42 eV correspond to the Pd 3d_3/2_ and Pd 3d_5/2_, which are associated with the metallic Pd [36]. The above results confirm that Pd was successfully deposited on the ZIS surface and existed in strong interactions.

### 2.2. Photocatalytic Performance

UV–vis DRS was applied to investigate the optical properties and band gaps of the synthesized samples. The absorption band edge of ZIS, as illustrated in Figure 4a, was approximately 536 nm, which corresponded to a band gap of 2.53 eV. The light absorption of Pd/ZIS in the visible area was improved when Pd was added. With an increase in the mass ratio of Pd in the composite, the adsorption intensity in the whole visible region gradually became stronger. The band gaps of ZIS and 1% Pd/ZIS were obtained from the Kubelka–Munk plots [50]. As displayed in Figure 4b, the band gaps of pure ZIS and 1% Pd/ZIS were calculated to be 2.53 and 2.48 eV, respectively. Moreover, the Mott–Schottky curves of ZIS exhibited positive slopes, suggesting that this semiconductor is an n-type one [51]. The flat-band potential of the sample was determined by extrapolating the lines to 1/C^2^ = 0 and was found to be −1.11 eV (vs. Ag/AgCl, pH = 7). According to the conversion relation between the Ag/AgCl electrode and the standard hydrogen electrode, E_NHE_ = E_Ag/AgCl_ + 0.197 [29], the CB position of the sample was finally calculated to be −0.91 eV (vs. NHE, pH = 6.8). Consequently, VB-XPS was able to identify the VB potential of ZIS. According to the VB-XPS plots of ZIS, the VB potential of ZIS was calculated to be 1.51 eV. The contact potential difference between the sample and the XPS analyzer was used to compute the VB potential of the normal hydrogen electrode (EVB-NHE vs. NHE, pH = 6.8) using the following calculation [24]:EVB-NHE = φ + EVB − XPS − 4.44 (1)

φ was the electron work function (4.55 eV) of the XPS analyzer, and EVB-XPS was the VB measured from the VB-XPS plots. Therefore, the EVB-NHE value of ZIS was calculated to be 1.62 eV. The band gap of ZIS could be computed using Mott–Schottky and VB-XPS spectral analyses, and it was matched with the bandgap values obtained from the UV–vis DRS spectra to obtain a value of 2.53 eV.

The photocatalytic activity of the samples was evaluated based on the photocatalytic degradation of pyridine under visible light (λ > 420 nm). As shown in Figure 5a, when the reaction system did not contain catalysts, the concentration of the pyridine solution did not decrease remarkably. Moreover, pure ZIS demonstrated the lowest activity, with only 36% pyridine degradation, which was ascribed to the rapid recombination of photogenerated carriers. However, after the Pd and CTAB loading, the activity of pyridine degradation significantly improved (Appendix A). Furthermore, the 1% Pd/ZIS composite exhibited the most effective photocatalytic activity and fuel denitrification efficiency of up to 81%. With the increase in the Pd loading ratio, the degradation rate of the Pd/ZIS decreased significantly, which was caused by the massive Pd load, resulting in the blocking of active sites. Moreover, we investigated the kinetics of the denitrification process in photocatalytic fuel using the pseudo-first-order model. As displayed in Figure 5d, the 1% Pd/ZIS photocatalyst exhibited the maximal rate constant (0.45 h^−1^), which was nearly 3.6 times greater than that of pristine ZIS (0.12 h^−1^). It was confirmed that there was a strong interaction between ZIS and Pd. Meanwhile, compared with the other catalysts, it was lower than that of the Pd/ZIS composite sample (Appendix A).

To explore the catalytic stability, we sought to detect the stability of 1% Pd/ZIS. As shown in Figure 6a, the fuel denitrification of the composite remained at a relatively high level. The degradation rate of pyridine still reached 73.5% after five cycles. Further research on the stability of 1% Pd/ZIS, XRD, and TEM was conducted to characterize the 5-cycle using 1% Pd/ZIS. From the XRD patterns (Figure 6b) and TEM images (Appendix A), no obvious change could be observed when comparing the used and fresh 1% Pd/ZIS, indicating that the composition and morphology of 1% Pd/ZIS were maintained after the reaction.

### 2.3. Adsorption Performance

According to the above experimental results for photocatalytic fuel denitrification, it is found that ZIS, Pd, and pyridine play vital roles in the photocatalytic performance. To further reveal the adsorption phenomenon of pyridine molecules on the Pd/ZIS surface, the adsorption behavior of pyridine molecules over the catalyst was investigated. As shown in Figure 7a (2), several characteristic peaks ascribed to pyridine (3080, 3037, 1582, and 1442 cm^−1^) could be observed after absorbing pyridine for 60 min. Meanwhile, these characteristic peaks could still be observed after vacuum degassing (Figure 7a (3)), indicating the strong chemisorption interaction between Pd/ZIS and pyridine molecules. The characteristic peaks at 3080 and 3037 cm^−1^ were attributed to the C-H stretching vibration of pyridine, while the characteristic peaks at 1582 and 1442 cm^−1^ were assigned to the stretching vibration of C-N and C-C [52]. Furthermore, with the evacuation at 150 °C, the characteristic peak of the stretching vibration of C-N at 1582 cm^−1^ showed an obvious shift, while the characteristic peak of C-H and C-C remained unchanged. It is speculated that the C-N groups of pyridine could be selectively chemisorbed on Pd/ZIS via the formation of surface coordination species to promote activity for fuel denitrification [53]. In addition, as shown in Figure 7b, the strong peaks at 1450 cm^−1^ and 1540 cm^−1^ corresponded to Lewis acids and Bronsted acids, revealing that the presence of abundant acid sites on Pd/ZIS and combined with pyridine (a Lewis base) [24]. Based on the above analysis, it can be concluded that Pd/ZIS has a strong binding affinity for pyridine.

To further study the interaction mechanisms between the representative pyridine molecules and the Pd/ZIS surface, we investigated the changes in elements’ (Zn, In, S, Pd) binding energy by XPS characterization of the ZIS and Pd/ZIS samples before and after pyridine adsorption. As demonstrated in Figure 8 and Appendix A, the peaks of Zn 2p were shifted to a low binding energy following pyridine adsorption in the XPS spectra of ZIS and Pd/ZIS samples, indicating that the electron cloud density of Zn was increased. It validated the production of Zn⋅⋅⋅C-N coordination between pyridine molecules and Pd/ZIS Zn^2+^ sites, leading to electron transfer from the C-N group to Zn^2+^ via Zn⋅⋅⋅C-N coordination [53]. Meanwhile, the binding energy shift of In 3d following pyridine adsorption was comparable to Zn 2p. These findings revealed that surface Zn^2+^ and In^3+^ would combine with C-N groups via Zn⋅⋅⋅C-N and In⋅⋅⋅C-N coordination, respectively. Notably, peaks of Zn 2p and In 3d of Pd/ZIS were higher than the shift of ZIS following pyridine adsorption, while 1% Pd/ZIS was the highest. It is proven that Pd loaded on the ZIS can lead to more electron transfer from the C-N group to Zn^2+^ and In^3+^ via Zn⋅⋅⋅C-N coordination and In⋅⋅⋅C-N coordination. It was further shown that the abundant surface Zn^2+^ and In^3+^ sites of Pd/ZIS as Lewis acid sites make a substantial contribution to the selective adsorption and activate C-N groups, significantly enhancing the pyridine selectivity. In addition, the UV–vis DRS spectra of Pd/ZIS adsorbed pyridine are displayed in Appendix A. The UV–vis DRS spectra of Pd/ZIS showed a redshift after adsorbing the pyridine molecule, which also indicated that surface coordination species were formed between the produced materials and pyridine [54].

### 2.4. Photocatalytic Mechanism

To explain further the efficiency of charge carrier transfer and separation, it was demonstrated by photoelectrochemical and photoluminescence (PL) studies. As shown in Figure 9a, the photocurrent intensity of Pd/ZIS was significantly higher than that of pure ZIS, while that of 1% Pd/ZIS was the highest, implying that 1% Pd/ZIS can more efficiently promote photogenerated carriers separation and transfer. Meanwhile, Figure 9b shows that the radius of 1% Pd/ZIS was the smallest, indicating that 1% Pd/ZIS can maximally reduce the interfacial charge transfer resistance. At the same time, according to the PL spectra of ZIS and Pd/ZIS, the PL intensity of 1% Pd/ZIS was the lowest, indicating that the photogenerated electrons of 1% Pd/ZIS can be more effectively separated [55]. From the above results, it is concluded that the Pd loaded on the ZIS can accelerate the separation and transfer of photogenerated carriers.

To provide a direct representation of the enhanced photocatalytic fuel denitrification after Pd loading, it was measured by HPLC-MS and ESR spectroscopy. The peak intensity of pyridine at around *m*/*z* = 80.1 was gradually decreased during the reaction process (Appendix A), indicating that the pyridine was successfully denitrogenated. Meanwhile, new peaks formed at *m*/*z* = 110.1, 116, and 61.1, revealing the conversion of pyridine to protonated intermediates such as C_4_H_4_O_3_, C_4_H_4_O_4_, and NH_2_COOH. Meanwhile, the pyridine adsorption value was obtained by conversion in Appendix A. The reactive species in the reaction system that the ESR detected should be further explored. As illustrated in Figure 10, TEMPO molecules were regarded as hole probes during the reaction process, because their free radicals could be oxidized by holes [56]. In the presence of 1% Pd/ZIS, the signal of TEMPO gradually decreases, revealing the generation of photogenerated holes (h^+^). With the extension of the illumination time, the signal of DMPO-•O_2_^−^ was enhanced step by step, which indicated that the •O_2_^−^ was the main active species during the reaction process. Meanwhile, the signal of DMPO-·OH was relatively weak, which was ascribed to the VB position of ZIS, which was lower than that of E (-OH/•OH; 1.99 eV) [57]. However, the generation of •OH radicals may originate from the •O_2_^−^ combined with H^+^ to form •OH [58].

Based on the above results, the photocatalytic mechanism (Figure 11, Appendix A) for the denitrification of pyridine over the Pd/ZIS composite was proposed [11]. When 1% Pd/ZIS composites are excited by visible light, the photogenerated electrons of Pd nanoparticles could migrate to the surface of the ZIS; once electrons reach ZIS, they react with pyridine and oxygen in the solution to yield a range of oxygen-containing products and •O_2_^−^, respectively. Meanwhile, the generated •O_2_^−^ reacts with H^+^ to form •OH radicals. The e^−^, •O_2_^−^, and •OH of reactive species could participate in the degradation of pyridine. The possible mechanism of denitrification can be found in the Appendix A. In the denitrification process, pyridine combines with the H^+^ and the •O_2_^−^ to generate a ring and a ring-opening intermediate product successively [11,59]. The intermediate product is converted into mineralization products including NO_3_^−^, CO_2_, and H_2_O in the presence of e^−^, •OH, and •O_2_^−^.

## 3. Materials and Methods

### 3.1. Materials

Zinc chloride (ZnCl_2_), indium chloride tetrahydrate (InCl_3_·4H_2_O), thioacetamide (TAA), cetyltrimethylammonium bromide (CTAB), palladium chloride (PdCl_2_), methanol, and ethanol were purchased from Sinopharm Chemical Reagent Co. Ltd.(Shanghai, China). and Macklin Reagent Co. Ltd. (Shanghai, China).

### 3.2. Synthesis of ZnIn_2_S_4_ Nanosheets (ZIS)

The ZIS was synthesized via a typical hydrothermal method [21]. First, 1 mmol ZnCl_2_, 2 mmol InCl_3_·4H_2_O, and 8 mmol TAA were dissolved in 30 mL of ethanol aqueous (ethanol:water = 1:2). Then, the mixture was transferred into a 50 mL Teflon-lined autoclave and heated at 180 °C for 2 h. After the reaction, the solid products were washed with deionized water and ethanol several times. Finally, the obtained products were dried at 60 °C for 12 h.

### 3.3. Synthesis of Pd/ZnIn_2_S_4_ Nanosheets (Pd/ZIS)

First, 100 mg ZIS was dispersed in a 20 mL aqueous solution containing 25 % methanol as a sacrificial agent. Pd (1%) was loaded on the ZIS surface using an in-situ photodeposition method with PdCl_2_ as a precursor (Figure 1). While under the N_2_ atmosphere, the mixture solution was stirred to remove O_2_. Afterward, the above solution was irradiated by a 300 W Xenon lamp for 1 h. Then, 70 mg CTAB was added to the above solution and stirred overnight. The final product was washed with ethanol and deionized water and collected by centrifugation and dried at 60 °C for 12 h.

### 3.4. Dispersion Experiment

A 5 mg sample was dissolved in 5 mL pyridine/octane solution (Appendix A).

### 3.5. Characterization Methods

The crystalline phase of the photocatalysts was obtained using a Bruker D8 Advance X-ray diffractometer (Salbruken, Germany). The surface morphology and microstructures were observed with SEM (Czech TESCAN MIRA LMS, Brno, Czech Republic) and TEM (FEI Talos F200s, Massachusetts, American) instruments. UV–vis absorption spectra were characterized by a Shimadzu UV-2700 (Kyoto, Japan), using BaSO4 as a reflectance standard. X-ray photoelectron spectroscopy (XPS) measurements were conducted on a Thermo Scientific K-Alpha+ spectrometer (Massachusetts, American). Photoluminescence (PL) spectra were characterized with a fluorescence spectrophotometer (Edinburgh FLS1000, Livingston, Scotland, UK) with an excitation wavelength of 225 nm. The in situ FTIR spectra of pyridine absorbed over the photocatalysts were collected on a Tensor 27 Fourier transform infrared (FT-IR) spectrometer (Salbruken, Germany) with a resolution of 4 cm^−1^ for 64 scans. EPR spectra of the prepared samples were obtained using a Germany Bruker EMX nano-spectrometer (Salbruken, Germany). Photoelectrochemical measurements were performed using an electrochemical workstation (CHI-660, Shanghai, China) with a conventional three-electrode cell, including the reference electrode (Pt wire), the counter electrode (Ag/AgCl), and the working electrode (FTO as a support). Mott–Schottky plots were generated with different frequencies of 500and 1000 Hz.

### 3.6. Performance Testing

The fuel denitrification of the photocatalyst was performed in a quartz reactor. First, 50 mL of pyridine/n-octane solution (70 mg/mL) was used to distribute 50 mg of photocatalyst. The solution was stirred for 1 h in the dark to achieve adsorption equilibrium. Then, using a 420 nm cutoff filter, the suspension was performed under 300 W Xe lamp irradiation (Appendix A). At given time intervals, 1.5 mL aliquots were sampled. A Varian Cary 50 spectrometer (Palo Alto, American) was used to monitor the residual concentration of pyridine in the supernatant at a 252 nm peak position.

## 4. Conclusions

In summary, we have successfully prepared a Pd/ZIS photocatalyst for the efficient photocatalytic denitrification of pyridine. When exposed to visible light, 1% Pd/ZIS produces a 2.25-fold greater reaction rate for the photocatalytic denitrification of pyridine than ZIS. From all the experimental results, the superior photocatalytic performance of 1% Pd/ZIS was ascribed to the enhanced visible light absorption, as well as the Zn⋅⋅⋅C-N or In⋅⋅⋅C-N coordination for pyridine molecules’ selective adsorption and activation. Meanwhile, the electron separation and transfer ability of Pd can be combined with ZIS to promote photocatalytic denitrification. This work provides a pathway for the photocatalytic denitrification of pyridine and explores organic molecules’ selective adsorption, activation, and photocatalytic conversion on metal/nanosheet surfaces.

## Data Availability

The data presented in the study are available from the corresponding author.

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
