# Peer review of "Surface Coordination of Pd/ZnIn2S4 toward Enhanced Photocatalytic Activity for Pyridine Denitrification"

_molecules, 2022, doi:10.3390/molecules28010282_

Round 1

Reviewer 1 Report

In the first place, it must be mentioned that the title is confusing since the term denitrification leads one to think of a reductive process since in the introduction the authors speak of eliminating the nitrogenous compounds present in the oil. The authors present evidence for the formation of oxidizing species but not for the possible reductive activity of the catalyst. The monitoring of the pyridine concentration in the photocatalytic processes only demonstrates its elimination but not the majority process. Certainly, it is possible to eliminate the nitrogen by oxidative route (after several stages) but then the oxidation of the non-nitrogenous compounds of the fuel mixtures can take place; something that would not be desirable.

It seems that it is convenient to clarify this concept to avoid misunderstandings and make the objective of the paper clear.

Reviewer 2 Report

I carefully read manuscript entitled “Surface coordination of ZnIn2S4 toward enhanced photocata-2 lytic for fuel denitrification”. The subject of the manuscript is interesting andworth of studing. The experiments are OK and well organized. However, the title is misleading a little bit. There is no mentioning of palladium at the surface of the ZIS and that is important if you talk about surface coordination. There is no explanation why pyridine is used as model compound for such complicated system as fuel – should we go thru literature to find the reasons? That is for the beginning.

Here my comments/recommendations for improval of the text:

1.       The title should be changed to include Pd on the surface of the photocatalyst

2.       Authors should explaine what is “surface coordination” of the ZIS. Usually word coordination is used for different crystalline phases and their different coordinations of dangling bonds at the surface which can induce different adsorption properties: physi and chemisorptions. I didn’t see any explanation of this in the manuscript.

3.       The term “surface plasmon resonance” is used only in teh abstract and abbreviation SPR in abstract and conclusion. There is no evidence of SPR in this Pd-ZIS system described in this manuscript. This should be deleted from the text.

4.       There is no explanation of effect Pd has on the photocatalysis except intensity of light absorption: XPS measurement reveald that pyridine is adsorbed at Zn and In sites at the surface of the ZIS. Is there any difference when pyridine is adsorbed on the ZIS when there is no Pd? You should measure XPS of the ZIS (without Pd) with pyridine as well. If you call that “surface coordination” it seems it is not changed with presence of Pd on the surface.

5.       The surface of the ZIS is probably highly covered with CTAB and Pd. Can you discuss the available free surface for photocatlytical reactions in that light/ regarding this fact as well.

6.       All other improved properties when Pd is present at the surface of ZIS (electrochemistry, photoinduced current) are interesting but of no significance for explanation of photocatalysis.

7.       The whole manuscript should be corrected for adsorption/absorption misspellings and wrongly used for different meanings.

After all correction manuscript should be reviewed again and considered for publication.

Best regards 

Reviewer 3 Report

The authors have well-established and presented the synthesis of Pd/ZIS composites for the photocatalytic degradation of pyridine. The title should be rewritten, though, adding pyridine and Pd words.. A missing word after "photocatalytic" is missing too.

The surface coordination, in my point of view, implies a systematic study of properly coordinating bonds between Pd and ZnIn2S4. This is missing from the work. Although there are experiments of XRD, UV-Vis absorbance and catalytic activity of different loadings of Pd content all the study is focused on the chosen catalysts with 1% wt loading of Pd. The coordination should be supported with further experiments of XPS and experiments reported in Figure 9 on other Pd contents in order to support a proper systematic study. Finally, ZnIn2S4 catalysts are reported in the title and not Pd/ZIS.

In the work, there is approximately 18/52 ~ 34 percent of self-citations. Authors are advised to enrich the literature due to the fact that the self-citation in the text is exceeding 30%. Please lower it below 28% by either enriching the literature or reducing the self-citations.

In the abstract, please include the Pd % of the and clarify (rephrase) the sentence in line 20.

In the introduction, a small paragraph is needed for the pyridine as a representative choice of petroleum nitrogen-containing compounds (NCCs).

line 54: improper use of the word "constructed". Please use "synthesized" or other similar

line 72: substitute the word "seen" with "observed".

Fig.2, Fig. 3, Fig.8 captions should refer to the Pd content

In Fig. 4a and Fig.4b please use different types of symbols or lines to distinguish the curves  (eg as in Fig.4c and Fig.5a)

In section 3.3 please refer to the amount of CTAB added to the synthesis route

Reviewer 4 Report

In the manuscript entitled “Surface coordination of ZnIn2S4 toward enhanced photocatalytic for fuel denitrification” degradation rate for fuel denitrification within 240 min, was investigated. The electrical conductivity of Pd, as well as the surface plasmon resonance (SPR) effect of Pd were explored.

The manuscript has proper structure and interesting subject. It can be published after revise the following issues:

1.      It is better to add a paragraph in introduction about importance of advanced oxidation processes and photocatalysis in fuel denitrification.

2.      What is the pyridin adsorption value?

3.      What is the test temperature? How authors control the temperature?

4.      According to EDS or XPS analysis, please report the experimental weight percent of Pd in samples.

Round 2

Reviewer 2 Report

Article can be accepted for publication now. Still English should be corrected and adsorption/absorption is somewhere misused.